# Self-Regulated Learning Strategies for Nursing Students: A Pilot Randomized Controlled Trial

**DOI:** 10.3390/ijerph19159058

**Published:** 2022-07-25

**Authors:** Jiwon An, Juyeon Oh, Kyongok Park

**Affiliations:** 1Department of Nursing, Far East University, Gamgok-myeon 27601, Korea; anjiwon@kdu.ac.kr; 2College of Nursing, Dankook University, Cheonan-si 31116, Korea; ohjy@dankook.ac.kr; 3Department of Nursing, Gangneung-Wonju National University, Wonju-si 26403, Korea

**Keywords:** self-regulated learning, augmented reality, perceived learning, knowledge, learning flow, academic stress, randomized controlled trial

## Abstract

Distance learning (DL) based on information and communication technologies is gaining importance due to its convenience and cost savings. However, there is not enough evidence to identify the effect of DL on students requiring a high level of self-regulated learning (SRL). Therefore, this study aims to compare the effects of the use of augmented reality (AR) as an innovative learning method and the use of a textbook as a conventional learning method. Both methods were based on SRL strategies. In this pilot randomized controlled trial (RCT), SRL using an AR group (*n* = 31) and a textbook group (*n* = 31) was performed. Perceived learning (PL) competency, knowledge, SRL competency, academic stress, and learning flow were measured to evaluate the effect of intervention. Although, there was not significant interaction between the effects of time and the intervention in PL competency, knowledge, academic stress, and learning flow. In the subdomains of SRL competency, environmental structuring, task strategies, time management, help seeking, and self-evaluation were significantly improved after intervention. SRL using innovative methods is more important after COVID 19. Therefore, well-designed larger RCTs are required to identify the effect of SRL strategy using innovative method.

## 1. Introduction

As a meta-cognitive learning strategy, self-regulated learning (SRL) allows learners to control and regulate their academic learning [1]. SRL also emphasizes the active role of learners and is strongly associated with learners’ motivation [2,3]. The educational environment is shifting from a teacher-oriented to a learner-oriented approach, and a learner-oriented environment requires SRL strategies [4]. COVID 19 changed everything, and the learning environment is not an exception. Most students were supposed to learn in physically distant digital classes and interacted with teachers using SNS, email, or intranet at school. Therefore, SRL has gained more interest after COVID 19 because of the social distance policy [5]. Previous studies reported the benefits and limitations of SRL. One study reported that students with higher SRL competency presented higher academic achievement [5]. The other study suggested that self-regulated learners constantly organize, monitor, and evaluate their study plans and eventually achieve better academic performance than non-self-regulated learners [6]. However, the method suitable for SRL—innovative or conventional—is still a topic of debate. Conventional education has been influenced by the 4th industrial revolution. This approach involves listening to a lecture while using books, and is no longer effective [7,8,9]. One study reported that traditional education is not suitable to apply SRL to as it requires a high level of student preparation as compared to innovative education methods, such as web-based or computer-based learning [3]. The study suggested that innovative education methods are more suitable for SRL because learners can control their own speed and learning process [5]. Another study reported that even though innovative education methods are more suitable for SRL, they require certain prerequisites to apply in learning. For example, an innovative environment such as a computer network or equipment, and the competency to handle the same is required [5]. Therefore, the effect of innovative education methods is limited when compared with conventional methods if the prerequisites required for the former are not fulfilled.

There are various technologically innovative educational methods such as the use of augmented reality (AR), virtual reality (VR), and haptic or hologram technology. These innovative methods allow students to learn from anywhere and at any time, contrary to the use of lectures with books [5,10,11]. Augmented reality has been applied in teaching for health care students [7,12,13]. AR, as a type of VR, is an interface technology that allows the user to a combination of virtual and real images. This technology can provide learners with advanced immersion and reality. AR bridges the virtual environment and real environment [14,15,16]. The main reason AR is used in learning is because it can provide a personalized and interactive learning experience. When compared to the conventional learning method, AR is able to instantly adapt to learners’ needs and give them feedback through real-time interaction [7,8]. AR can increase learners’ achievement and satisfaction because it can present and visualize complicated concepts using a three-dimensional space, and the learner can control the learning content and his or her learning process [8]. The learners’ ability to control the learning process using AR makes it a suitable tool to help learners study individually.

Furthermore, AR is receiving attention because of its capability of solving faculty shortages or providing an alternative for the use of limited onsite laboratory space [17]. Currently, students are called ‘digital natives,’ indicating that it is much easier and more comfortable for them to communicate with others and study in the digital world rather than simply listen and observe [12]. Therefore, learning and teaching strategies should reflect the digital orientation of learners to enhance their educational effects.

Anatomy is a necessary course for those striving to become nurses who need to learn basic biological sciences, such as physiology, pathology and pharmacology [18]. There are many traditional methods to learn anatomy, such as the use of cadavers, models, videos, and books. Learning anatomy using cadavers has been practiced for a long time and is known to be very practical and effective, but also very costly [13]. It is difficult for undergraduate first- or second-year nursing students to learn anatomy with a cadaver because of traumatic emotional disturbance or exposure to toxic chemical substances [19]. Learning anatomy using models with almost the same shape and appearance as a real human body, or videos, is relatively inexpensive but is limited in the presentation of spatial and physical characteristics with three-dimensional images [7]. Learning anatomy through AR can be relatively lower in cost and can present three-dimensional images. Hence, the use of AR to teach anatomy has been increasingly prominent in health care education, such as in the medical and nursing fields, and the effects have been studied [20]. Although there have been many studies on the effects of AR on learning, studies such as usability studies to test a developed prototype [12] and observational studies [13] have limitations in identifying the effects. A few studies have reported that AR increases self-confidence [21], knowledge [22] and learning flow [23]. However, there has been no study examining an SRL strategy on learning with AR. In order to measure the effectiveness of SRL strategy, the most important variable is SRL competency. According to previous studies, learning competency, knowledge, learning flow, and academic stress are important variables to evaluate the effectiveness of SRL strategies [5,24,25,26]. Some of the educational benefits of using AR in learning reported in previous studies were: an increase in learning competency, learning flow, and a decrease in stress, directly or indirectly [27]. This study is a pilot study designed as a randomized controlled trial aimed at testing the effectiveness of SRL based on AR technology as an innovative learning method for nursing students as compared to the effectiveness of conventional learning with the use of a textbook.

## 2. Materials and Methods

### 2.1. Design

This pilot study was conducted to develop strategies for larger RCTs to identify the effect of SRL strategies using innovative methods. It was a two-arm, randomized, parallel group trial. Two groups were allocated at a 1:1 ratio to identify the effects of SRL on the AR group compared with the textbook group.

### 2.2. Participants

Participants were first- and second-year nursing students from two universities in Korea. The university has approximately 75 students per grade, and the anatomy class in the regular curriculum is held in the second semester of the first year. After the approval of the Institutional Review Board of the University, the author’s affiliation (GWNUIRB-2021-07), potential participants were voluntarily recruited through an intranet notice at the university. The participants were informed of the purpose and procedures of the study, and then voluntarily completed written informed consent. Sixty-two participants who met the eligibility criteria were enrolled and randomized into two groups. The eligibility criteria for the participants were as follows: (a) first- and second-year students in the nursing department; (b) those who owned and used smartphones; (c) those who understood the purpose and content of the trial and voluntarily agreed to participate; and (d) those who had not participated in other anatomy learning studies during the semester.

### 2.3. Interventions

#### 2.3.1. Materials

The textbook used in this trial was developed by the authors. This textbook was used for SRL in both the groups. This textbook consisted of the following two contents: (a) the skeletal system, at a moderate level; and (b) the structure of the heart, at a high level. The textbook contained learning objectives, study guides and brief descriptions, such as the anatomical location, name, and function; it was provided to all participants in the AR and textbook groups.

An AR mobile application (DEVAR Entertainment LLC, Marlton, NJ, USA) with image-based AR was provided to the participants in the AR group. The 3D virtual anatomical objects were shown in the AR book, and the participants could display the objects on the mobile screen with the use of a marker on the page of the AR book using the camera in the mobile app. The AR app, which operated on iPhone and/or Android platforms, displayed realistic three-dimensional (3D) images of the human body. Participants were able to manipulate the 3D object to explore it from different perspectives; they could drag the image to rotate it 360° and tap the icons on the mobile screen to show the anatomical terms, internal structures and disassembled images of organs (Figure 1). The illustrations and the trial code for the AR app were provided free of charge by the app company for research purposes [28,29]. Participants in the experimental group performed four weeks of SRL using the AR app along with the textbook. The experiment was conducted from 3 to 31 May 2021. In the control group, participants were provided with the textbook without the AR app, and they performed four weeks of SRL with the textbook only.

#### 2.3.2. Procedure

Before starting the intervention, participants were informed of the purpose and process of the trial, and were encouraged by the researcher to study anatomy. Participants from both groups were given 15 min to complete the pretest to assess their SRL competency, perceived learning (PL) competency, knowledge, academic stress, learning flow and demographic information. The participants in the experimental group were given an additional five-minute explanation about how to use the AR app. During the experimental period, participants learned anatomy using the AR app or the textbook only for four weeks. To increase adherence to the procedure, the researcher, who was blinded from the group allocation, contacted participants to check their learning progress using Zoom online meeting once a week and conducted a question-and-answer session. The researcher explained SRL strategies such as goal setting, environment structuring, task strategies, time management, help seeking, and self-evaluation in the first Zoom online meeting. After that, from the second to fifth Zoom online meetings, the researcher encouraged them to use the strategies, and answered questions about learning methods, such as how to use the textbook or how to use the AR app. However, the researcher did not give lectures on the learning content so as not to affect outcome variables. As a final step, all participants were given 15 min to complete a post-test to assess outcome variables similar to the pre-test. For ethical reasons, participants in the control group were given the opportunity to try the AR app after the study was finished. The control participants’ use of the AR app was not included in the trial.

### 2.4. Outcome Measures

The outcomes were measured with self-reported questionnaires pre- and post-experiment by a blinded researcher.

The demographic data collected from the participants included age, grade level, grade point average (GPA), perceived academic achievement, augmented reality experience, and Zoom meeting attendance.

SRL competency was measured using the Online Self-Regulated Learning Questionnaire (OSRLQ), a 24-item questionnaire with a 5 point Likert scale [9]. It consists of six sub-scales, including goal setting, environment structuring, task strategies, time management, help seeking, and self-evaluation. Higher scores indicate a higher degree of SRL. The Cronbach’s alpha of the OLSQ was 0.90 in the original study and 0.89 in this study.

PL competency in this study refers to the perceived cognitive, affective, and psychomotor effects of SRL. The CAP Perceived Learning Scale [30], a nine-item, 6 point Likert scale, was used to measure PL competency. The score ranged from 0 to 54, with a higher score indicating higher perceptions of learning. The Cronbach’s alpha of the CAP Perceived Learning Scale was 0.79 in the original study and 0.73 in this study.

Knowledge was assessed with a 10-item quiz that included questions based on the textbook content. Participants in both groups completed the 10-item quiz during the pre- and post-test. The quiz consisted of multiple-choice questions about the skeletal system and heart. The quiz items were reviewed by a nursing professor with teaching experience in anatomy considering the validity, accuracy and relevance of the study content. Each question was scored 1 for the correct answer and 0 for incorrect answers. The total score ranged from 0 to 10, with a higher score representing a higher level of knowledge. The Kuder–Richardson (KR)-20 was 0.76.

Learning flow refers to the degree to which participants experienced immersion in anatomy learning, and was measured using the short version of the Flow State Scale developed by Jackson and colleagues [31]. Flow state, a positive experiential state, is defined as the moment when the performer is totally connected to the performance in a variety of situations [32]. The short version of the Flow State Scale has nine items that assess challenge–skill balance, action–awareness merging, clear goals, unambiguous feedback, concentration on the task at hand, sense of control, loss of self-consciousness, transformation of time, and autotelic experience. Item scores are measured on a 5 point Likert scale where a higher score indicates a higher level of learning flow. The Cronbach’s alpha of the short version of the Flow State Scale was 0.82 in the original study and 0.81 in this study.

Academic stress was assessed with the Perception of Academic Stress (PAS) Scale, an 18 item with a 5 point Likert scale [33]. It contains the stresses related to students’ academic self-perceptions, faculty work and examinations, and academic expectations. A higher score indicates less academic stress. The Cronbach’s alpha of the PAS was 0.70 in the original study and 0.78 in this study.

### 2.5. Sample Size

The sample size was calculated using the G *Power program (Version 3.1.9.2, Franz Faul, University Kiel, Kiel, Germany). To achieve the desired effect size, an effect size of 0.67 [34] was applied in the sample size calculation. Seventy-two participants were needed to achieve a power of 0.80 given the two measurement times (α = 0.05). This study included 80 participants to allow for a 10% dropout rate.

### 2.6. Randomization

Participants were randomly allocated to the AR group or the textbook group in a 1:1 ratio. The randomization was performed by an independent researcher using a computer-generated random number system. Participants were not informed of their assigned groups until the experiment was initiated.

### 2.7. Statistical Analysis

Statistical analyses were conducted using SPSS, version 22 (IBM Corp., Somers, NY, USA). The Shapiro-Wilk test, skewness, and kurtosis tests were used to evaluate the normal distribution of the variables. The homogeneity of participants’ characteristics and outcome variables at the pretest were analyzed using descriptive statistics, independent t-tests, chi-square tests, and Fisher’s exact test. Comparisons of differences in SRL competency, PL competency, knowledge, learning flow and academic stress between the two groups were analyzed by the repeated measures analysis of variance (ANOVA). Differences between pre-test and post-test was analyzed by paired t-test. Statistical significance was determined as a two-tailed *p* value below 0.05.

### 2.8. Ethical Considerations

This study was approved by the institutional review board of the author’s university (GWNUIRB-2021-07). The study was conducted independently from the regular curriculum, and the authors did not have any role in the participants’ evaluation as part of the regular curriculum. The authors provided detailed information about the purpose and procedures of the study to the participants, ensured the anonymity of participants’ personal information and collected data, and used a separate study ID for each participant; data management was performed only by the third researcher. Before the participants provided written consent, they were informed that there was no penalty for not participating and that they could withdraw at any time during the study. All participants were given approximately $10 as compensation for their participation in the study, and the participants in the control group were given the opportunity to use the AR app after completion of the trial.

## 3. Results

A flow chart presenting the enrollment, experiment, random allocation, follow-up, and data analysis of the study is shown in Figure 2. Recruitment and follow-up were conducted from 1 to 31 May 2021. 72 participants were recruited. Participants filled out their informed consent after sufficient explanation about the study. Excluding the five participants who denied filling out informed consent, 67 participants were enrolled and randomly assigned to the two groups. Three participants from experimental and two participants from control group were dropped after randomization. The study was finished when participants completed the follow-up assessment after receiving their assigned interventions.

### 3.1. Participant’s Characteristics and Homogeneity Test

There were 31 participants in each of the two groups. The characteristics of the participants and the results of the homogeneity tests are shown in Table 1. The mean age of the participants was 20.2 ± 2.9. There were 33 (53.2%) first-year students and 29 (46.8%) second-year students. The mean GPA was 3.9 ± 0.4 based on 4.5 scale. 82.3% of participants answered. In the zoom meeting held during the experiment period, 34 participants attended 0–2 times, and 28 participants attended 3–5 times. There were no significant differences in characteristics between the two groups.

### 3.2. Comparison of Intervention Effect

The comparison of the effect of intervention between the groups and time was analyzed by repeated measures ANOVA. There was not significant interaction between the effects of time and the intervention in PL competency, knowledge, academic stress, and learning flow. SRL competency in the textbook group significantly increased than in the AR group after intervention (F (1,60) = 5.68, *p* = 0.020) (Table 2).

The result of the difference after intervention presented that SRL competency (F = 18.07, *p* < 0.001), PL competency (F =28.02, *p* < 0.001), knowledge (F = 37.30, *p* < 0.001), and learning flow (F = 8.85, *p* = 0.004) were increased significantly regardless of groups. There was no significant increase in academic stress.

### 3.3. The Difference of SRL Competency between Pre-Post Test

The effect of SRL components was analyzed by paired t-test, as shown in Table 3. As subdomains of SRL competency, environmental structuring (t = −2.83, *p* = 0.006), task strategies (t = −2.20, *p* = 0.032), time management (t = −3.28, *p* = 0.002), help seeking (t = −3.04, *p* = 0.003), and self-evaluation (t = −3.96, *p* < 0.001) were significantly improved after intervention. There was no significant improvement in goal setting.

Additionally, participants asked a lot of questions regarding SRL strategies during a question-and-answer session. For example, how to memorize efficiently, time management for assignment during clinical practice, health condition management strategies, and even role models they strived to be like and their relationship with friends or professors. Therefore, participants not only wanted information about learning strategies but also their self-confidence or career path in the future.

## 4. Discussion

This study compared the effectiveness of SRL strategies between the AR group as an innovative educational method and the textbook group as a conventional method for nursing students. This pilot study was conducted to develop strategies for larger RCTs. SRL-competency improved more in the textbook group after four weeks of the learning period. This result implied that the use of innovative educational technology was not a superior method to improve SRL competency among nursing students as compared to the conventional method in the present study. The authors elucidated why the use of AR was not a superior method to improve SRL competency for nursing students. First, the study period was not long enough to improve SRL competency. Second, applying innovative methods in learning required an innovative environment, such as possession of a smartphone, free Wi-Fi, or internet speed as compared to the environment for textbook usage [35,36]. Third, the level of technology for the AR program was not as high as expected by participants. Although current studies focus on evaluating prototypes, the use of AR in nursing may have positive implications [37]. Therefore, future studies should focus on performing long-term evaluations of AR.

There are many pieces of evidence showing that innovative learning method is superior to conventional learning. The study reported that SRL competency in computer-based learning groups made students feel autonomous and motivated [38]. Another study reported that technology-enhanced learning environments led to the successful improvement of SRL strategies and eventually improved students’ academic achievements [39]. Zimmerman [40] suggested that the systematic use of motivational and behavioral strategies to optimize learning is a key feature of self-regulated learners. Zimmerman & Pons [41] reported that seeking information, environmental structuring, rehearsing, and memorizing to make learning easier were important components of SRL. Muali et al. [24] reported that students with higher SRL-competency using mobile AR and conventional learning methods showed a higher conceptual understanding [24]. Other studies reported that students using AR gained learning confidence as a foundation of SRL strategy [21]. In particular, a previous literature review reported that the benefits of AR included an increase in motivational learning, self-learning, and independence in learning and eventually led to improved academic performance [27]. Barmaki et al. [23] conducted a quasi-experimental study and reported that anatomical learning using AR improved the retention of knowledge as opposed to conventional learning with a textbook. Researchers studied the impact of a learning method that involved overlaying anatomical visualizations on students’ bodies using AR magic mirrors on the knowledge of anatomy among medical students compared to that of a learning method with passive textbooks. They reported that AR-based learning contributed to increasing knowledge levels by enhancing students’ participation. A few studies supported these findings, reporting improvements in knowledge, academic achievement, and cognitive abilities due to AR-based anatomy learning [11,22].

In this present study, SRL-competency was measured with goal setting, environmental structuring, task strategies, time management, help seeking, and self-evaluation. All components of SRL, except for goal setting, increased after the intervention. One study reported instructors can use knowledge about SRL in various ways, including discussion of the benefit of SRL, open-ended instructional activity with students, and minimizing competitive test scores, when they apply the SRL strategies in the classroom [42]. In terms of effectiveness in the AR group, a visual image that looked similar to a real patient was viewed using Google Glass and a wearable head device was provided for AR learning. The author reported that simulations based on AR technology increased nursing students’ perception of reality and consequently enhanced students’ learning confidence in a clinical setting [21]. In particular, help seeking, one of the SRL components, improved after the intervention. Generally, this implied that different strategies from traditional learning are required because SRL-based technology using AR, VR, or the internet created a physical distance between learners and instructors. Garcia et al. [32] reported that help seeking is difficult because students tend to obtain the answer from peers and not from teachers in a digital environment. Therefore, instructors should try to connect with learners through familiar ways such as email, SNS, or the intranet of the school at any time or at any physical space. In addition, instructors could use the help seeking chance to discuss with students about learning questions for contents or objectives as well as learning strategies, learning barriers, learning goals, or even learners’ careers in the future. The role of a coach is also important for instructors in an SRL environment because instructors may easily communicate with learners through email, SNS, or the intranet of school, and might be preferred by learners as one of their closed-one. In the present study, students had a meeting with a research assistant once a week to improve SRL competency, wherein students asked the research assistant about time management including the arrangement of sleeping time and learning time, learning strategies about how to memorize, management of career in the future, and self-confidence. It was an important chance to increase the effect of SRL competency.

In this study, the knowledge score, perceived learning, and learning flow improved significantly after the intervention in both groups, even though there was no difference between groups. Previous studies reported that SRL affected academic performance, including knowledge as well as cognitive, affective, and psychomotor domains, and learning flow [5,43]. SRL strategy is based on behavioral self-regulation, metacognitive self-monitoring, and environmental structuring to adapt thinking skills to various educational situations [31,32]. After COVID-19, distance learning gained popularity as a learning method, thus garnering more interest in SRL strategies [5].

Generally, learning methods based on technology such as AR, VR, or simulation, expect students to increase learning flow [5,44,45,46]. In the present study, learning flow improved in both groups after intervention. Previous studies reported that students became more interested in learning through attractive images and showed higher immersion through direct manipulation experiences. Another study demonstrated that technology-enhanced learning tools, such as 3D models or VR and AR solutions enhance students’ engagement in the context of anatomy education [23,47,48]. Subjects that require high levels of learning engagement and are challenging, such as anatomy, the use of educational technologies can be a new strategy to facilitate SRL by inducing learners to immerse and repeat learning [17,49]. In addition, learning flow has a relationship with the learning environment, thus the instructors coached them to choose an appropriate time and place to focus on the learning to improve academic performance. However, academic stress did not decrease significantly after the intervention. The present study evaluated academic stress not only for self-perception of learning but also for academic expectations and faculty work and examinations. The intervention period was only four weeks. This short intervention period is the reason why it did not demonstrate a significant change in academic stress.

Recently, research on the effects of anatomical learning using AR or VR is being actively conducted [50,51,52]. Previous studies reported that the educational effect of AR is that learners can study and train safely and repeatedly, regardless of location or time [53,54]. This suggests that AR technology could be a good strategy for improving the effectiveness of self-regulated learning. In addition, these technologies can be used to improve learner-centered education in various educational fields. Learner-centered education is not limited to time and place and can introduce a dynamic and interactive relationship between learning experiences and outcomes to improve the quality and effectiveness of nursing education [55,56]. To achieve a high level of satisfaction, educational content development should include content that is interactive and focused rather than using traditional educational approaches [57]. Therefore, developing apps that include appropriate levels of content and interactions, such as step-by-step pop messages or quizzes, could be a strategy to further enhance the effectiveness of self-regulated learning.

This was a pilot study to determine whether SRL with AR or textbook is feasible for nursing students in learning anatomy. There are some limitations. The AR method was not enough to identify the effect on academic competency in this result because of the small sample size, short intervention period, random enrollment of participants from two universities, and AR apps with lower technology than participants’ expectations. In addition, it does not meet the sample size calculated because the intervention period coincided with the examination period of some nursing students. Therefore, it is necessary to conduct larger RCTs to test the effectiveness of SRL using efficient innovative educational technology for an entire semester. This study only investigated nursing students in the context of anatomy learning; therefore, SRL interventions must be applied to various subjects such as fundamental nursing or adult nursing. It is also required that more in-depth studies using qualitative and quantitative approaches will be beneficial to understand more about SRL ability of nursing students. Despite these limitations, this pilot RCT found that learning anatomy by applying SRL strategy with AR helped participants improve their SRL competency.

## 5. Conclusions

This study was conducted to identify the effectiveness of SRL between two groups of nursing students: one studying with an innovative learning method such as AR technology; and the other studying with a conventional learning method such as using a textbook. Environmental structuring, task strategies, time management, help seeking, and self-evaluation among the subdomains of SRL-competency were significantly improved after intervention. SRL-competency using innovative technology is becoming more important because of the demand for contactless education, such as that required for the prevention of COVID-19. Maintaining SRL-competency is not easy for learners because it requires them to independently seek information, organize and transform, and seek social assistance from peers and teachers. However, using information and communication technology can make learners feel comfortable by providing suitable information and access assistance with ease, thereby leading to successful SRL. In the future, these education strategies based on SRL and the use of innovative technology will lead to higher academic achievement and be a universal education method that is favored over conventional education approaches, such as face-to-face lectures.

## Figures and Tables

**Figure 1 ijerph-19-09058-f001:**
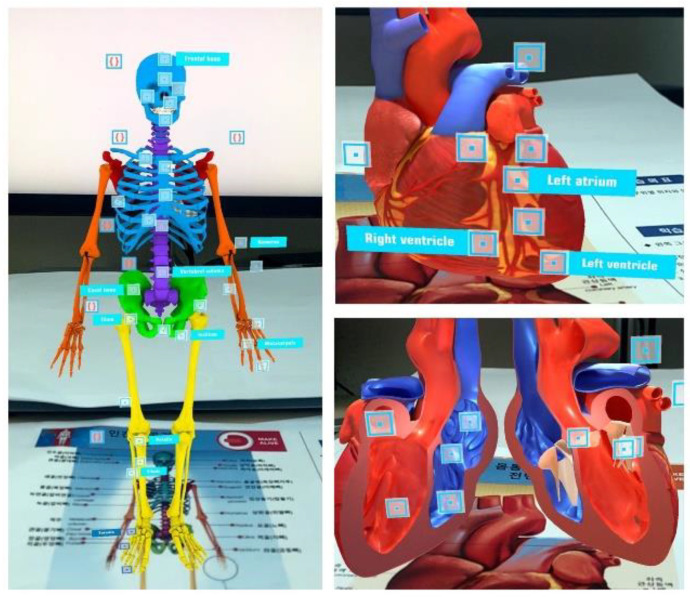
Screenshot from the Augmented Reality Application.

**Figure 2 ijerph-19-09058-f002:**
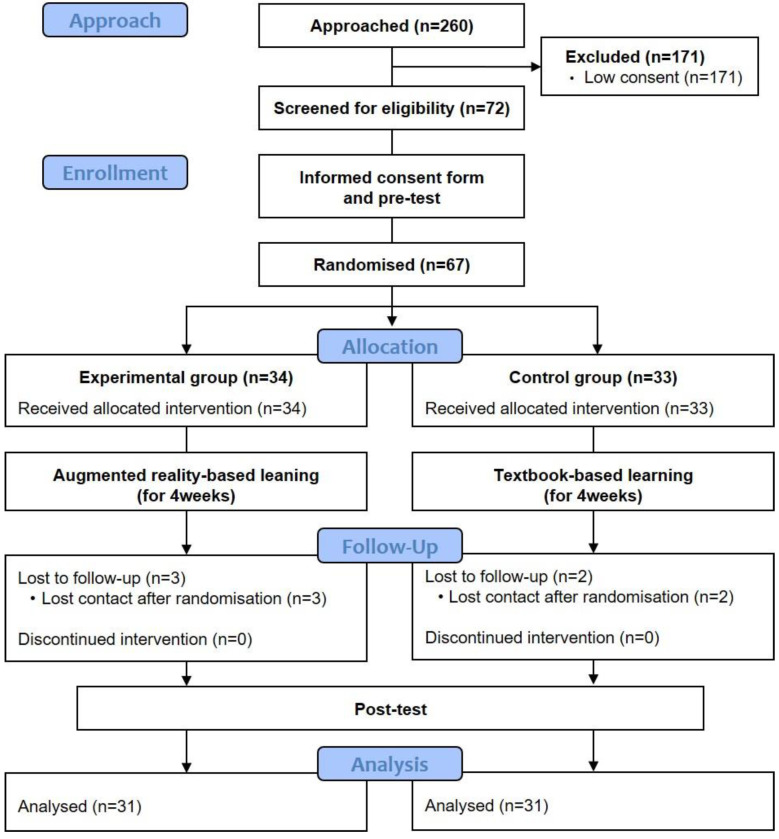
Study Flow Chart.

**Table 1 ijerph-19-09058-t001:** Homogeneity for Demographic Characteristics between Two Groups (*n* = 62).

Characteristics	Categories	Total (*n* = 62)	AR Group (*n* = 31)	Textbook Group (*n* = 31)	χ^2^ or t	*p*
		n (%) or M ± SD				
Age (yr)		20.2 ± 2.9	20.3 ± 3.0	20.2 ± 2.8	0.03	0.863
Grade level	Freshmen	33 (53.2)	16 (51.6)	17 (54.8)	0.07	0.801
	Sophomore	29 (46.8)	15 (48.4)	14 (45.2)		
Grade point average (4.5 scale)		3.9 ± 0.4	3.8 ± 0.4	3.9 ± 0.4	0.66	0.418
Subjective academic achievement	High	7 (11.3)	4 (12.9)	3 (9.7)	1.31	0.629 *
	Moderate	51 (82.3)	24 (77.4)	27 (87.1)		
	Low	4 (6.5)	3 (9.7)	1 (3.2)		
Augmented reality experience	Yes	11 (17.7)	4 (12.9)	7 (22.6)	1.00	0.324
	No	51 (82.3)	27 (87.1)	24 (77.4)		
Frequency of Zoom Meeting Attendances	0–2	34 (54.8)	18 (58.1)	16 (51.6)	0.80	0.401
	3–5	28 (45.2)	13 (41.9)	15 (48.4)		

AR: augmented reality; * Fisher’s exact test.

**Table 2 ijerph-19-09058-t002:** Comparison of intervention effect between Groups (*n* = 62).

Variables	AR Group (*n* = 31)	Textbook Group (*n* = 31)	Source	F	*p* *
	Mean ± SD				
SRL competency					
Pre-test	3.5 ± 0.6	3.5 ± 0.6	Group	0.73	0.397
Post-test	3.7 ± 0.4	3.9 ± 0.5	Time	18.07	<0.001
			Group X Time	5.68	0.020
PL competency					
Pre-test	33.1 ± 9.0	32.8 ± 6.6	Group	0.10	0.751
Post-test	38.7 ± 7.4	38.0 ± 6.2	Time	28.02	<0.001
			Group X Time	0.04	0.837
Knowledge					
Pre-test	7.3 ± 2.3	7.8 ± 2.2	Group	0.98	0.327
Post-test	9.0 ± 1.1	9.2 ± 1.2	Time	37.30	<0.001
			Group X Time	0.34	0.565
Learning flow					
Pre-test	3.3 ± 0.7	3.2 ± 0.6	Group	0.00	0.957
Post-test	3.5 ± 0.6	3.6 ± 0.6	Time	8.85	0.004
			Group X Time	2.68	0.107
Academic stress					
Pre-test	3.3 ± 0.5	3.3 ± 0.5	Group	0.61	0.438
Post-test	3.4 ± 0.5	3.3 ± 0.5	Time	0.02	0.883
			Group X Time	0.13	0.724

PL: perceived learning; SRL: self-regulated learning; * Greenhouse-Geisser.

**Table 3 ijerph-19-09058-t003:** Mean difference of SRL competency between pretest and posttest (*n* = 62).

Variables	Pretest	Posttest	t *	*p*
	Mean ± SD			
SRL competency	3.5 ± 0.6	3.8 ± 0.5	−4.10	<0.001
Goal setting	3.8 ± 0.7	3.9 ± 0.6	−1.07	0.290
Environmental structuring	4.1 ± 0.7	4.3 ± 0.5	−2.83	0.006
Task strategies	3.2 ± 0.8	3.4 ± 0.6	−2.20	0.032
Time management	3.5 ± 0.8	3.8 ± 0.6	−3.28	0.002
Help seeking	3.3 ± 0.9	3.5 ± 0.8	−3.04	0.003
Self-evaluation	3.3 ± 0.8	3.6 ± 0.8	−3.96	<0.001

SRL: self-regulated learning; * Paired *t*-test.

## Data Availability

Data available on request due to restrictions e.g., privacy or ethical.

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
