# Peer review of "Self-Regulated Learning Strategies for Nursing Students: A Pilot Randomized Controlled Trial"

_ijerph, 2022, doi:10.3390/ijerph19159058_

Round 1
Reviewer 1 Report
General comments
This study aimed to compare the effectiveness of SRL strategy for nursing students between use of AR technology as an innovative learning method and the conventional learning with the use of a textbook. This study assumed that AR technology supports attractive and engaging learning materials by promoting the self-regulated learning. However, the AR method was not enough to identify the more effective method to improve SRL competency in present results. Although the hypothesis was not support, the concept of this study is still valuable for more understanding of the SRL strategy for nursing students. However, the recently relative literatures are insufficient, only one paper after 2020. The relative studies such as the effect of AR and SRL, SRL in nursing education and AR in nursing education are needed. Some lasted studies for your reference.
1. Quqandi, E., Joy, M., Drumm, I., & Rushton, M. (2022). Augmented Reality in Supporting Healthcare and Nursing Independent Learning: Narrative Review. CIN: Computers, Informatics, Nursing, 10-1097.
2. Chytas D, Johnson EO, Piagkou M, et al. The role of augmented reality in anatomical education: an overview. Annals of Anatomy. 2020;229: 151463.
3. Muali, C., Setyosari, P., Purnomo, P., & Yuliati, L. (2020). Effects of Mobile Augmented Reality and Self-Regulated Learning on Students’ Concept Understanding. International Journal of Emerging Technologies in Learning (iJET), 15(22), 218-229.
4. Wüller, H., Behrens, J., Garthaus, M., Marquard, S., & Remmers, H. (2019). A scoping review of augmented reality in nursing. BMC nursing, 18(1), 1-11.
The specific comments:
Introduction
In Line 55 to Line 56, the explanation of “innovative education methods” should be more specifically. The suggestion is “There are various technological based innovative educational methods such as using of augmented reality (AR), virtual reality (VR), and haptic or hologram technology.”
Method
In the outcome measurement part authors mentioned the dependent various such as SRL competency, Perceived learning (PL) competency, knowledge, academic stress, learning flow. Besides SRL competency was mentioned, why measured Perceived learning (PL) competency, knowledge, academic stress, learning flow should be also explained.
Discussion
There was a lack of studies integrating AR and self-learning theories (such as self-regulated learning). This pilot study determined whether SRL with AR or textbook is feasible for nursing students in learning anatomy. The discussion part could be having more discussion about why SRL competency increased after the intervention in both group and the specific effect of AR group.
Author Response
|
Thank you for your comments! We have revised the manuscript based on the reviewer’s comments. We have the English editing according to your comments about English language and style.
1. Introduction : In Line 55 to Line 56, the explanation of “innovative education methods” should be more specifically. The suggestion is “There are various technological based innovative educational methods such as using of augmented reality (AR), virtual reality (VR), and haptic or hologram technology.” Response : Thank you for your comments. I have revised the explanation of “innovative education methods” as per your suggestion in lines 53–54.
2. Method : In the outcome measurement part authors mentioned the dependent various such as SRL competency, Perceived learning (PL) competency, knowledge, academic stress, learning flow. Besides SRL competency was mentioned, why measured Perceived learning (PL) competency, knowledge, academic stress, learning flow should be also explained. Response : Thank you for the comments. The aim of the study is to identify the effectiveness of SRL strategy using educational methods such as innovative and conventional methods. In order to measure the effectiveness of SRL strategy, the most important variable is SRL competency. However, learning competency (PL, GPA, and knowledge), learning flow, and academic stress are also important variables to evaluate the effectiveness of SRL strategy. In a previous study [5], SRL competency was correlated with learning competency, learning flow, and engagement. Although the relationship between SRL competency and academic stress is not supported in the previous study, the relationship between academic achievement and stress is well known [53–54]. In particular, the educational benefit of AR in learning also includes an increase in learning competency and learning flow and a decrease in stress, directly or indirectly, as reported in a previous study [55]. I have summarized the reason why SRL competency, PL competency, knowledge, academic stress, and learning flow were measured as dependent variables in lines 93–99.
3. Discussion : There was a lack of studies integrating AR and self-learning theories (such as self-regulated learning). This pilot study determined whether SRL with AR or textbook is feasible for nursing students in learning anatomy. The discussion part could be having more discussion about why SRL competency increased after the intervention in both group and the specific effect of AR group. |
Response : While conducting this study, we doubted whether the assumption for the benefit of AR was correct because of the lack of references. However, we were confident that an innovative educational method, such as AR, is a great solution, especially in an environment requiring distance learning and SRL, such as that during COVID 19. I appreciate the help with the references. I have added an explanation for the reason why SRL competency increased after the intervention in both groups and the specific effect of the AR group in lines 325–348. In addition, we described a question-and-answer session in the Results section. Prior to engaging in these experiments, the students already knew the importance of SRL strategy in distance learning amid COVID 19, but they did not know how to apply the SRL strategy. Consequently, they asked numerous questions at every session. We have discussed this in the Discussion section in line 371-375.

Reviewer 2 Report
How to improve nursing students' self-regulated learning ability is an important issue in digital learning environment especially in a time of COVID-19 pandemic. The paper showed the result of a pilot randomized controlled trial using a AR group and a textbook group conducted in South Korea.
More attention could be paid to the external factors (e.g. real practice environment, teachers, etc.) and internal factors (e.g. curiosity, awareness, etc.) that could improve their SRL ability. In future, more in-depth studies that using qualitative and quantitative approaches will be beneficial to understand more about SRL ability of nursing students.
Author Response
|
How to improve nursing students' self-regulated learning ability is an important issue in digital learning environment especially in a time of COVID-19 pandemic. The paper showed the result of a pilot randomized controlled trial using a AR group and a textbook group conducted in South Korea. More attention could be paid to the external factors (e.g. real practice environment, teachers, etc.) and internal factors (e.g. curiosity, awareness, etc.) that could improve their SRL ability. In future, more in-depth studies that using qualitative and quantitative approaches will be beneficial to understand more about SRL ability of nursing students. |
Response : Thank you for your comments! We have revised the manuscript based on the reviewer’s comments. We have the English editing according to your comments about English language and style. I have added your opinion in the Discussion section in lines 427–429.

Reviewer 3 Report
The article is very interesting. Investigating innovative teaching techniques for nursing students is highly pertinent.
The introduction frames the subject of study very well and the methodology used is impeccable.
The results are presented clearly.
In the discussion, by expressing that the hypothesis raised in the study is not verified, the authors expose their assumptions about the causes of this fact. I suggest that you omit these comments from the discussion and include them in the study limitations section.
In my opinion, the conclusions do not reflect the main findings of the results, it gives the impression that they have been written prior to the end of the study according to the results that they expected to have and have not had. I suggest that you modify and adapt them to the results of this research.
Author Response
|
1. The article is very interesting. Investigating innovative teaching techniques for nursing students is highly pertinent. The introduction frames the subject of study very well and the methodology used is impeccable. The results are presented clearly. In the discussion, by expressing that the hypothesis raised in the study is not verified, the authors expose their assumptions about the causes of this fact. I suggest that you omit these comments from the discussion and include them in the study limitations section. |
Response : Thank you for your comments! We have revised the manuscript based on the reviewer’s comments. We have the English editing according to your comments about English language and style. I have deleted the sentences from lines 316 and 317 accordingly. I have also incorporated it in the Limitation section in lines 317–321.
|
2. In my opinion, the conclusions do not reflect the main findings of the results, it gives the impression that they have been written prior to the end of the study according to the results that they expected to have and have not had. I suggest that you modify and adapt them to the results of this research. |
Response : Thank you for your comments. I have revised the Conclusion section, lines 433–438.

Round 2
Reviewer 1 Report
This study is valuable for more understanding about SRL competency using innovative technology for distance learning. The revised manuscript and responses provided sufficient explanations for all the comments.